# A Cancer-Specific Monoclonal Antibody against Podocalyxin Exerted Antitumor Activities in Pancreatic Cancer Xenografts

**DOI:** 10.3390/ijms25010161

**Published:** 2023-12-21

**Authors:** Hiroyuki Suzuki, Tomokazu Ohishi, Tomohiro Tanaka, Mika K. Kaneko, Yukinari Kato

**Affiliations:** 1Department of Molecular Pharmacology, Tohoku University Graduate School of Medicine, 2-1 Seiryo-machi, Aoba-ku, Sendai 980-8575, Japan; tomohiro.tanaka.b5@tohoku.ac.jp (T.T.); mika.kaneko.d4@tohoku.ac.jp (M.K.K.); 2Department of Antibody Drug Development, Tohoku University Graduate School of Medicine, 2-1 Seiryo-machi, Aoba-ku, Sendai 980-8575, Japan; 3Institute of Microbial Chemistry (BIKAKEN), Microbial Chemistry Research Foundation, 18-24 Miyamoto, Numazu-shi 410-0301, Japan; ohishit@bikaken.or.jp; 4Institute of Microbial Chemistry (BIKAKEN), Laboratory of Oncology, Microbial Chemistry Research Foundation, 3-14-23 Kamiosaki, Shinagawa-ku, Tokyo 141-0021, Japan

**Keywords:** podocalyxin, PODXL, cancer-specific monoclonal antibody, defucosylated antibody, pancreatic cancer

## Abstract

Podocalyxin (PODXL) overexpression is associated with poor clinical outcomes in various tumors. PODXL is involved in tumor malignant progression through the promotion of invasiveness and metastasis. Therefore, PODXL is considered a promising target of monoclonal antibody (mAb)-based therapy. However, PODXL also plays an essential role in normal cells, such as vascular and lymphatic endothelial cells. Therefore, cancer specificity or selectivity is required to reduce adverse effects on normal cells. Here, we developed an anti-PODXL cancer-specific mAb (CasMab), PcMab-6 (IgG_1_, kappa), by immunizing mice with a soluble PODXL ectodomain derived from a glioblastoma LN229 cell. PcMab-6 reacted with the PODXL-positive LN229 cells but not with PODXL-knockout LN229 cells in flow cytometry. Importantly, PcMab-6 recognized pancreatic ductal adenocarcinoma (PDAC) cell lines (MIA PaCa-2, Capan-2, and PK-45H) but did not react with normal lymphatic endothelial cells (LECs). In contrast, one of the non-CasMabs, PcMab-47, showed high reactivity to both the PDAC cell lines and LECs. Next, we engineered PcMab-6 into a mouse IgG_2a_-type (PcMab-6-mG_2a_) and a humanized IgG_1_-type (humPcMab-6) mAb and further produced the core fucose-deficient types (PcMab-6-mG_2a_-f and humPcMab-6-f, respectively) to potentiate the antibody-dependent cellular cytotoxicity (ADCC). Both PcMab-6-mG_2a_-f and humPcMab-6-f exerted ADCC and complement-dependent cellular cytotoxicity in the presence of effector cells and complements, respectively. In the PDAC xenograft model, both PcMab-6-mG_2a_-f and humPcMab-6-f exhibited potent antitumor effects. These results indicated that humPcMab-6-f could apply to antibody-based therapy against PODXL-expressing pancreatic cancers.

## 1. Introduction

Podocalyxin (PODXL) is a CD34-related transmembrane sialomucin glycoprotein [1]. The 53-kDa PODXL core protein undergoes extensive *N*- and *O*-linked glycosylation, leading to a mature protein with a molecular weight of 150,000–200,000 [2]. PODXL is normally expressed by embryonic stem cells [3], early hematopoietic progenitors [4], adult vascular/lymphatic endothelial cells [5], and kidney podocytes [6]. PODXL plays a critical role in cancer development, and PODXL-knockout (KO) mice display lethality before birth [7]. Importantly, PODXL is aberrantly expressed in a broad range of human tumors, including oral squamous cell carcinoma [8], colorectal cancer [9], renal cell carcinoma [10], and pancreatic ductal adenocarcinoma (PDAC) [11]. Increased PODXL expression can have adverse effects on overall survival, disease-specific survival, and disease-free survival in several cancers [12].

PODXL regulates epithelial and tumor cell motility through interactions with the actin polymerization complexes, including the ERM (ezrin–radixin–moesin) family [13] and PDZ protein Na^+^/H^+^ exchanger regulatory factors 1 and 2 (NHERF-1/2) [14]. Ezrin and NHERF-1/2 are adaptor proteins that have diverse binding partners and facilitate the interaction of PODXL with the cytoskeletons of tumor cells [12]. The interactions between PODXL, ezrin, and NHERF-1/2 have been reported to activate intracellular signaling including Rac1, RhoA, Cdc42, mitogen-activated protein kinase, and phosphatidylinositol-3 kinase pathways to promote motility [12]. PODXL has been shown in gain- and loss-of-function studies to play an essential role in tumor progression by promoting migration, invasiveness, stemness, and metastasis in a variety of cancer cells [15]. Therefore, PODXL has acquired increasing attention as a target of tumor immunotherapy.

Preclinical studies have shown the promising efficacy of anti-PODXL monoclonal antibodies (mAbs). A core protein-binding anti-PODXL mAb (PODO83/PODOC1) suppressed MDA-MB-231 xenograft growth and blocked metastasis to the lung [16]. Our group also established an anti-PODXL mAb, PcMab-47 (mouse IgG_1_, kappa), which is useful for flow cytometry and immunohistochemistry [17]. We engineered PcMab-47 into 47-mG_2a_, a mouse IgG_2a_-type mAb, to add antibody-dependent cellular cytotoxicity (ADCC) and further developed 47-mG_2a_-f, a core fucose-deficient type of 47-mG_2a_ to augment its ADCC. Both 47-mG_2a_ and 47-mG_2a_-f showed antitumor activity against PODXL-expressing oral squamous cell carcinoma-bearing mouse models [18]. Furthermore, we developed a mouse–human chimeric PcMab-47, chPcMab-47, and showed its antitumor efficacy against PODXL-positive colorectal adenocarcinomas [19].

Nevertheless, the further development of anti-PODXL mAbs targeting tumors has been hampered by concerns about possible toxicities to normal vascular/lymphatic endothelial cells [5] and kidney podocytes [6]. For the development of antibody therapy against PODXL-positive cancers, cancer-specificity is necessary to reduce the risk of adverse effects on normal tissues. In this study, we developed a cancer-specific anti-PODXL mAb, PcMab-6 (IgG_1_, kappa), by screening more than one hundred hybridoma clones. We engineered PcMab-6 into a mouse IgG_2a_-type (PcMab-6-mG_2a_) and a humanized IgG_1_-type (humPcMab-6) mAb and further produced the core fucose-deficient types (PcMab-6-mG_2a_-f and humPcMab-6-f, respectively) to potentiate ADCC. This technique was clinically applied to mogamulizumab (Poteligeo^®^), a defucosylated antibody targeting CCR4 [20]. We then examined the antitumor activity against mouse xenograft models of PDAC.

## 2. Results

### 2.1. Development of an Anti-PODXL CasMab, PcMab-6

Using recombinant PODXL derived from glioblastoma LN229 as an antigen, we developed more than one hundred clones of anti-PODXL mAbs. We further screened the reactivity to cancer and normal cells compared with a control mAb, PcMab-47 [17], using flow cytometry. As shown in Figure 1A, a novel anti-PODXL mAb, PcMab-6 (IgG_1_, kappa), reacted with endogenous PODXL on a glioblastoma LN229 cell line but not with PODXL-KO LN229 (PDIS-13). In contrast, PcMab-47 showed stronger reactivity to LN229 but not to PDIS-13 (Figure 1A). Similar reactivities were also observed in PDAC cell lines, including MIA PaCa-2, Capan-2, and PK-45H (Figure 1B). PODXL is highly expressed in vascular and lymphatic endothelial cells [5]. As shown in Figure 1C, PcMab-47 stained the PODXL of a lymphatic endothelial cell (LEC), HDMVEC/TERT164-B. In contrast, PcMab-6 did not react with it (Figure 1C), indicating that PcMab-6 shows cancer-specific reactivity.

### 2.2. Production of PcMab-6-mG_2a_-f, a Defucosylated Mouse IgG_2a_-Type PcMab-6

We engineered a mouse IgG_2a_-type PcMab-6 (PcMab-6-mG_2a_) by fusing the V_H_ chain of PcMab-6 with the C_H_ chain of mouse IgG_2a_ (Figure 2A). We further produced the core-fucose-deleted version, PcMab-6-mG_2a_-f, using Fut8-deficient ExpiCHO-S (BINDS-09) cells (Figure 2A). We then investigated whether PcMab-6-mG_2a_-f could exert ADCC against pancreatic cancer cells in the presence of BALB/c nude mice-derived splenocytes as an effector. As shown in Figure 2B, PcMab-6-mG_2a_-f showed ADCC against MIA PaCa-2 (66.0% vs. 24.2% cytotoxicity of control mIgG_2a_, *p* < 0.01), Capan-2 (78.9% vs. 32.2% cytotoxicity of control mIgG_2a_, *p* < 0.05), and PK-45H (57.2% vs. 41.3% cytotoxicity of control mIgG_2a_, *p* < 0.05). Control mIgG_2a_ is a reference mAb that does not recognize PODXL.

We then examined whether PcMab-6-mG_2a_-f could exhibit complement-dependent cellular cytotoxicity (CDC) against pancreatic cancer cells. As shown in Figure 2C, PcMab-6-mG_2a_-f elicited a higher degree of CDC against MIA PaCa-2 (82.9% vs. 14.8% cytotoxicity of control mIgG_2a_, *p* < 0.01), Capan-2 (74.4% vs. 49.0% cytotoxicity of control mIgG_2a_, *p* < 0.05), and PK-45H (64.0% vs. 35.5% cytotoxicity of control mIgG_2a_, *p* < 0.01). These results demonstrated that PcMab-6-mG_2a_-f exerted higher levels of ADCC and CDC against pancreatic cancer cells, which depend on the recognition of PODXL.

### 2.3. Antitumor Activity of PcMab-6-mG_2a_-f against Pancreatic Cancer Xenografts

Following the inoculation of MIA PaCa-2 or Capan-2, PcMab-6-mG_2a_-f or control mIgG was intraperitoneally injected into pancreatic cancer xenograft-bearing mice on days 7, 14, and 21. On days 7, 9, 14, 16, 21, 23, and 28 after the inoculation, the tumor volume was measured. The PcMab-6-mG_2a_-f administration resulted in a significant reduction in MIA PaCa-2 xenografts on days 16 (*p* < 0.01), 21 (*p* < 0.01), 23 (*p* < 0.01), and 28 (*p* < 0.01) compared with that of control mIgG (Figure 3A). A significant reduction was also observed in the Capan-2 xenograft on days 21 (*p* < 0.05), 23 (*p* < 0.01), and 28 (*p* < 0.01) (Figure 3B).

In the PK-45H xenograft, PcMab-6-mG_2a_-f or control mIgG was intraperitoneally injected into the mice on days 4, 11, and 18. On days 4, 6, 11, 13, 18, 20, and 25 after the inoculation, the tumor volume was measured. The PcMab-6-mG_2a_-f administration resulted in a significant reduction in PK-45H xenografts on days 11 (*p* < 0.05), 13 (*p* < 0.01), 18 (*p* < 0.01), 20 (*p* < 0.01), and 25 (*p* < 0.01) compared with that of control mIgG (Figure 3C).

A significant reduction in xenograft weight caused by PcMab-6-mG_2a_-f was observed in MIA PaCa-2 (60% reduction; *p* < 0.01; Figure 3D), Capan-2 (33% reduction; *p* < 0.05; Figure 3E), and PK-45H (27% reduction; *p* < 0.05; Figure 3F). The MIA PaCa-2 and Capan-2 xenografts that were resected from mice on day 28 are demonstrated in Figure 3G,H, respectively. The PK-45H xenografts that were resected from mice on day 25 are demonstrated in Figure 3I.

Body weight loss and skin disorders were not observed in the xenograft-bearing mice (Figure 3J–L). The mice on day 28 (MIA PaCa-2 and Capan-2) and day 25 (PK-45H) are shown in Appendix A.

### 2.4. Production of a Humanized and Defucosylated Antibody, humPcMab-6-f

We next engineered a humanized PcMab-6 (humPcMab-6) by fusing the V_H_ and V_L_ complementarity determining regions (CDRs) of PcMab-6 with the C_H_ and C_L_ chains of human IgG_1_, respectively (Figure 4A). We further produced the core-fucose-deficient version of humPcMab-6 (humPcMab-6-f), as described above, and investigated whether humPcMab-6-f can detect PODXL-expressed cancer cells. The humPcMab-6-f showed moderate reactivity to LN229 (Figure 4B) and PDAC cell lines, such as MIA PaCa-2, Capan-2, and PK-45H (Figure 4C), and showed very low reactivity to HDMVEC/TERT164-B (Figure 4D). The humPcMab-6-f did not recognize PODXL-KO LN229 (PDIS-13), as shown in Figure 4B.

### 2.5. ADCC and CDC Caused by humPcMab-6-f against Pancreatic Cancers

We next examined ADCC caused by humPcMab-6-f against pancreatic cancer cells in the presence of human natural killer (NK) cells as an effector. As shown in Figure 5A, humPcMab-6-f showed ADCC against MIA PaCa-2 (61.5% vs. 13.6% cytotoxicity of control hIgG, *p* < 0.05), Capan-2 (56.2% vs. 16.2% cytotoxicity of control hIgG, *p* < 0.01), and PK-45H (72.9% vs. 16.7% cytotoxicity of control hIgG, *p* < 0.01). Control hIgG is a reference mAb that does not recognize PODXL.

We then investigated CDC caused by humPcMab-6-f against pancreatic cancer cells. As shown in Figure 5B, humPcMab-6-f elicited a higher degree of CDC against MIA PaCa-2 (74.7% vs. 39.6% cytotoxicity of control hIgG, *p* < 0.05), Capan-2 (46.3% vs. 7.3% cytotoxicity of control hIgG, *p* < 0.01), and PK-45H (60.8% vs. 13.3% cytotoxicity of control hIgG, *p* < 0.05). These results demonstrated that humPcMab-6-f exerted very high levels of ADCC and CDC against pancreatic cancer cells, which depend on the recognition of PODXL.

### 2.6. Antitumor Activity of humPcMab-6-f against Pancreatic Cancer Xenografts

Following the inoculation of the pancreatic cancers, humPcMab-6-f or control hIgG was intraperitoneally injected into pancreatic cancer xenograft tumor-bearing mice on days 7 and 14. Human NK cells were also injected around the tumors on days 7 and 14. On days 7, 10, 14, 17, and 21 after the inoculation, the tumor volume was measured. The humPcMab-6-f administration resulted in a significant reduction in MIA PaCa-2 xenografts on days 10 (*p* < 0.05), 14 (*p* < 0.01), 17 (*p* < 0.01), and 21 (*p* < 0.01) compared with that of hIgG (Figure 6A). A significant reduction was also observed in the Capan-2 xenograft on days 17 (*p* < 0.05) and 21 (*p* < 0.01) (Figure 6B) and in PK-45H xenograft on days 10 (*p* < 0.01), 14 (*p* < 0.01), 17 (*p* < 0.01), and 21 (*p* < 0.01) (Figure 6C).

A significant reduction in xenograft weight caused by humPcMab-6-f was observed in MIA PaCa-2 (30% reduction; *p* < 0.05; Figure 6D), Capan-2 (31% reduction; *p* < 0.01; Figure 6E), and PK-45H (26% reduction; *p* < 0.05; Figure 6F). The MIA PaCa-2, Capan-2, and PK-45H xenografts that were resected from mice on day 21 are demonstrated in Figure 6G,H,I, respectively.

Body weight loss and skin disorders were not observed in the xenograft-bearing mice (Figure 6J–L). The mice on day 21 are shown in Appendix A.

## 3. Discussion

Pancreatic cancer has become the third leading cause of death in men and women combined in the United States as of 2023 [21]. PDAC is the most common type of pancreatic cancer and exhibits an extremely poor prognosis with a 5-year survival rate of approximately 10% [22]. The four most common oncogenic events, including *KRAS*, *CDKN2A*, *SMAD4*, and *TP53*, play critical roles in cancer development [23,24]. In contrast, PDAC is a heterogenous tumor with various histologies [25], heterogenous molecular landscapes [26], and clinical outcomes. Therefore, the identification of early diagnostic markers and therapeutic targets in each group is desired. In this study, we provided evidence that PODXL could be a promising target for antibody-based therapy, especially PcMab-6 (Figure 1) and its derivatives, including PcMab-6-mG_2a_-f (Figure 2 and Figure 3) and humPcMab-6-f (Figure 4, Figure 5 and Figure 6).

High PODXL expression is significantly associated with worse overall survival and is predictive of shorter overall survival in multiple cancers, especially PDAC [27]. Furthermore, PODXL is upregulated during epithelial–mesenchymal transition [28] and plays a key role in the extravasation of mesenchymal PDAC cells [29]. PODXL enhances the extravasation through direct binding to ezrin, a cytoskeletal linker protein, which enables the transition of tumor cells from a non-polarized, rounded cell shape to an invasive extravasation-competent morphology [29]. These results suggest that PODXL mediates the process of extravasation during tumor metastasis. Although the requirement of PODXL ligands such as E-selectin [30] in endothelial cells is unknown, it would be interesting to see whether PcMab-6 affects PODXL ligand interactions, tumor cell extravasation, and metastasis.

We developed a PODXL-targeting CasMab in cancer cells by screening more than one hundred clones of anti-PODXL mAbs. PcMab-6 can recognize PDAC cells but not normal LECs in flow cytometry (Figure 1). We previously developed CasMabs against podoplanin (PDPN) [31]. The CasMabs were selected via flow cytometry and immunohistochemistry and exhibited reactivity with cancer cells but reduced reactivity with normal cells [32]. An anti-PDPN CasMab (clone LpMab-2), which has been applied to chimeric antigen receptor (CAR)-T therapy in mice preclinical models [33], recognizes the glycopeptide from Thr55 to Leu64 of PDPN [32]. Another CasMab, LpMab-23, recognizes a peptide from Gly54 to Leu64 of PDPN [34].

The minimum epitopes of one of the anti-PODXL non-CasMabs (clone PcMab-47) and another anti-PODXL CasMab (clone PcMab-60 [35]) were determined to be the peptide sequence, including Asp207, His208, Leu209, and Met210, of PODXL [36] and the peptide sequence, including Arg109, Gly110, Gly111, Gly112, Ser113, Gly114, Asn115, and Pro116, of PODXL [37], respectively. In contrast, the epitope of PcMab-6 has not been identified yet. The identification of the PcMab-6 epitope is essential for understanding the mechanism of cancer-specific recognition. Furthermore, recognition by PcMab-6 in flow cytometry was lower than that of PcMab-47 (Figure 1), which can be attributed to the possibility that the epitope sequence is partially exposed or modified in cancer cells but not in normal cells in living cells. The CasMab selection strategy can contribute to the development of novel mAbs against a variety of antigens.

Glycosylation in tumors is frequently altered [38]. Anti-PODXL mAbs, which recognize tumor-specific glyco-epitopes on PODXL but do not react with PODXL expressed on normal cells, have been developed [39]. One of these mAbs, PODO447, exhibits exquisite specificity for a tumor glycoform of PODXL but lacks reactivity with normal adult human tissue. Using an array of glycosylation defective cell lines, the epitope of PODO447 was revealed to be an *O*-linked core 1 glycan presented in the context of the PODXL peptide backbone [39]. Furthermore, PODO447 coupled with monomethyl auristatin E (MMAE) showed efficacy in targeting human pancreatic and ovarian tumor xenografts in mouse models [40].

All therapeutic mAbs exhibit adverse effects, probably because of the recognition of antigens expressed in normal cells. The majority of patients treated with anti-EGFR therapeutic mAbs experience dermatological disorders, which affect their quality of life and adherence to the therapy [41]. The major adverse effect associated with anti-HER2 therapeutic mAbs is cardiotoxicity [42]. Since PODXL is expressed in normal adult tissues, including vascular/lymphatic endothelial cells [5] and kidney podocytes [6], the evaluation of the in vivo toxicity of PcMab-6 derivatives is essential. We investigated this toxicity against cynomolgus monkeys. The first protocol is one intravenous injection of 20 mg/kg of the mouse–human chimeric and core-fucose-deleted antibody of PcMab-6 (chPcMab-6-f) and observation for one week. The second protocol is four intravenous injections of 10 mg/kg of chPcMab-6-f (every week) and observation for one month. We confirmed that chPcMab-6-f did not show any toxicities against cynomolgus monkeys in either protocol (Appendix A).

CARs are synthetic modular proteins that redirect immune cell reactivity toward a target of interest. This platform has shown substantial clinical effects against B cell and plasma cell malignancies and the potential to expand its application to solid tumors [43]. The optimal epitope for the PODXL-targeting CAR has not yet been evaluated. However, it would be worthwhile to investigate the cancer specificity of the PcMab-6 single-chain variable fragment and the efficacy of CAR-T therapy against PODXL-positive tumors.

## 4. Materials and Methods

### 4.1. Cell Lines

A mouse myeloma cell line, P3X63Ag8U.1 (P3U1); a human glioblastoma cell line (LN229); and a PDAC cell line (Capan-2) were obtained from the American Type Culture Collection (Manassas, VA, USA). Two PDAC cell lines (MIA PaCa-2 and PK-45H) were obtained from the Cell Resource Center for the Biomedical Research Institute of Development, Aging, and Cancer of Tohoku University (Miyagi, Japan). The LN229/PODXL ectodomain (LN229/PODXLec) was produced in our previous study [17]. PODXL-KO LN229 (PDIS-13) was generated as described previously [19]. LN229, PDIS-13, LN229/PODXLec, and MIA PaCa-2 were cultured in Dulbecco’s modified Eagle’s medium (DMEM) (Nacalai Tesque, Inc., Kyoto, Japan). Capan-2 was cultured in McCoy’s 5A medium (Cytiva, Tokyo, Japan). P3U1 and PK-45H were cultured in Roswell Park Memorial Institute (RPMI)-1640 medium (Nacalai Tesque, Inc., Kyoto, Japan). All media were supplemented with 10% heat-inactivated fetal bovine serum (FBS; Thermo Fisher Scientific Inc., Waltham, MA, USA), 100 units/mL of penicillin, 100 μg/mL of streptomycin, and 0.25 μg/mL of amphotericin B (Nacalai Tesque, Inc.). A lymphatic endothelial cell line (HDMVEC/TERT164-B) was purchased from EVERCYTE (Vienna, Austria) and was cultured in an Endopan MV kit (PAN Biotech, Bayern, Germany) supplemented with G418. All cells were cultured at 37 °C in a humidified atmosphere containing 5% CO_2_ and 95% air.

### 4.2. Animals

For the establishment of mAbs against PODXL, the animal experiment was approved by the Animal Care and Use Committee of Tohoku University. For the ADCC assay and antitumor activity in mouse xenograft models, the animal experiments were approved by the Institutional Committee for Experiments of the Institute of Microbial Chemistry. The loss of original body weight to a point > 25% and/or a maximum tumor size > 3000 mm^3^ were identified as humane endpoints for euthanasia.

### 4.3. Hybridoma Production

Female BALB/c mice (CLEA, Tokyo, Japan) were immunized with purified PODXLec (100 μg) [17] together with Imject Alum (Thermo Fisher Scientific Inc.) via intraperitoneal injection. After several additional immunizations of PODXLec, the spleen cells were harvested and fused with P3U1 cells using PEG1500 (Roche Diagnostics, Indianapolis, IN, USA). Hybridomas were grown in RPMI-1640 medium, including L-glutamine with hypoxanthine, aminopterin, and thymidine (HAT) selection medium supplement (Thermo Fisher Scientific Inc.). Culture supernatants were screened using enzyme-linked immunosorbent assay (ELISA) for binding to PODXLec.

### 4.4. ELISA

PODXLec was immobilized on Nunc Maxisorp 96-well immunoplates (Thermo Fisher Scientific, Inc.) at 1 μg/mL for 30 min. After blocking with 1% bovine serum albumin (BSA) in 0.05% Tween 20/phosphate-buffered saline (PBS, Nacalai Tesque, Inc.), the plates were incubated with culture supernatant followed by 1:2000 diluted peroxidase-conjugated anti-mouse immunoglobulins (Agilent Technologies, Inc., Santa Clara, CA, USA). The enzymatic reaction was produced with a 1-Step Ultra TMB-ELISA (Thermo Fisher Scientific, Inc.). The optical density was measured at 655 nm using an iMark microplate reader (Bio-Rad Laboratories, Inc., Berkeley, CA, USA).

### 4.5. Antibodies

A mouse anti-PODXL mAb, PcMab-47 (IgG_1_, kappa), was developed as described previously [17]. Normal mouse IgG (mIgG) was purchased from FUJIFILM Wako Pure Chemical Corporation (Osaka, Japan). Mouse IgG_2a_ (mIgG_2a_) and normal human IgG (hIgG) were purchased from Sigma-Aldrich Corp. (St. Louis, MO, USA). To generate PcMab-6-mG_2a_, the V_H_ cDNA of PcMab-6 and the C_H_ of mouse IgG_2a_ were subcloned into a pCAG-Ble vector (FUJIFILM Wako Pure Chemical Corporation), and the V_L_ and C_L_ cDNAs of PcMab-6 were subcloned into a pCAG-Neo vector (FUJIFILM Wako Pure Chemical Corporation). To generate a humanized PcMab-6 (humPcMab-6), the CDR of PcMab-6 V_H_, the frame sequences of V_H_ in human Ig, and the C_H_ of human IgG_1_ were cloned into the pCAG-Neo vector. The CDR of PcMab-6 V_L_, the frame sequences of V_L_ in human Ig, and the C_L_ of the human kappa chain were cloned into the pCAG-Ble vector. To generate PcMab-6-mG_2a_-f and humPcMab-6-f, antibody expression vectors were also transfected into BINDS-09 (Fut8-knocked-out ExpiCHO-S cells) using the ExpiCHO Expression System. PcMab-6, PcMab-6-mG_2a_-f, and humPcMab-6-f were purified using Protein G-Sepharose (GE Healthcare Bio-Sciences, Pittsburgh, PA, USA).

### 4.6. Flow Cytometry

Cell lines were harvested via brief exposure to 0.25% trypsin/1 mM ethylenediaminetetraacetic acid (EDTA; Nacalai Tesque, Inc.). After washing with 0.1% BSA in PBS (blocking buffer), the cells were treated with primary mAbs for 30 min at 4 ◦C, followed by treatment with Alexa Fluor 488-conjugated anti-mouse IgG (1:1000; Cell Signaling Technology, Danvers, MA, USA) or fluorescein isothiocyanate (FITC)-conjugated anti-human IgG (1:2000; Sigma-Aldrich Corp.). Fluorescence data were collected using an SA3800 Cell Analyzer (Sony Corp., Tokyo, Japan).

### 4.7. ADCC

The ADCC activity of PcMab-6-mG_2a_-f was measured as follows. In brief, effector cells were obtained from the spleens of female BALB/c nude mice (Jackson Laboratory Japan, Inc., Kanagawa, Japan). We labeled target cells (MIA PaCa-2, Capan-2, and PK-45H) using 10 µg/mL of Calcein AM (Thermo Fisher Scientific, Inc.), plated them on 96-well plates (1 × 10^4^ cells/well), and mixed them with the effector cells (effector-to-target ratio, 100:1) with 100 μg/mL of control mIgG_2a_ or PcMab-6-mG_2a_-f. After a 4.5 h incubation at 37 °C, the Calcein release into the medium was measured using a microplate reader (Power Scan HT; BioTek Instruments, Inc., Winooski, VT, USA).

The ADCC activity of humPcMab-6-f was measured as follows. The Calcein AM-labeled target cells (MIA PaCa-2, Capan-2, and PK-45H) were mixed with human NK cells (Takara Bio, Inc., Shiga, Japan; effector-to-target ratio, 50:1) with 100 μg/mL of control hIgG or humPcMab-6-f. The Calcein release into the medium was measured after a 4.5 h incubation.

Cytolyticity (% lysis) was determined: % lysis is calculated as (E − S)/(M − S) × 100, where “E” denotes the fluorescence in the effector and target cell cultures, “S” denotes the spontaneous fluorescence of only target cells, and “M” denotes the maximum fluorescence after treatment with a lysis buffer (10 mM Tris-HCl (pH 7.4), 10 mM EDTA, and 0.5% Triton X-100). All data are shown as mean ± standard error of the mean (SEM). Welch’s *t*-test was used for the statistical analyses.

### 4.8. CDC

The Calcein AM-labeled target cells (MIA PaCa-2, Capan-2, and PK-45H) were plated and mixed with rabbit complement (final dilution 1:10, Low-Tox-M Rabbit Complement; Cedarlane Laboratories, Hornby, ON, Canada) and 100 μg/mL of control mIgG_2a_, PcMab-6-mG_2a_-f, control hIgG, or humPcMab-6-f. Following incubation for 4.5 h at 37 °C, the Calcein release into the medium was measured, as described above.

### 4.9. Antitumor Activity of Anti-PODXL Antibodies

Female BALB/c nude mice were purchased from Jackson Laboratory Japan, Inc. The cells (0.3 mL of 1.33 × 10^8^/mL in DMEM) were mixed with 0.5 mL of BD Matrigel Matrix Growth Factor Reduced (BD Biosciences, San Jose, CA, USA). A 100 μL suspension (containing 5 × 10^6^ cells) was injected subcutaneously into the left flanks of the nude mice. In total, 100 μg of PcMab-6-mG_2a_-f or control mIgG in 100 μL of PBS was injected into the peritoneal cavity of each mouse at day 4 (PK-45H) or day 7 (MIA PaCa-2 and Capan-2). Additional antibodies were injected on days 11 and 18 (PK-45H) or days 14 and 21 (MIA PaCa-2 and Capan-2). The mice were euthanized on day 25 (PK-45H) or day 28 (MIA PaCa-2 and Capan-2) after the cell inoculation.

To measure the antitumor activity of humPcMab-6-f, 100 μg of humPcMab-6-f or control hIgG in 100 μL of PBS was injected into the peritoneal cavity of each tumor xenograft-bearing mouse on day 7. Additional antibodies were injected on day 14. Furthermore, human NK cells (8.0 × 10^5^ cells) were injected around the tumors on days 7 and 14. The mice were euthanized on day 21 after the cell inoculation.

The tumor diameter was measured and the tumor volume was calculated using the following formula: volume = W^2^ × L/2, where W is the short diameter, and L is the long diameter. All data are expressed as the mean ± SEM. Statistical analysis was performed using ANOVA with Sidak’s post hoc test. *p* < 0.05 was considered statistically significant.

## Figures and Tables

**Figure 1 ijms-25-00161-f001:**
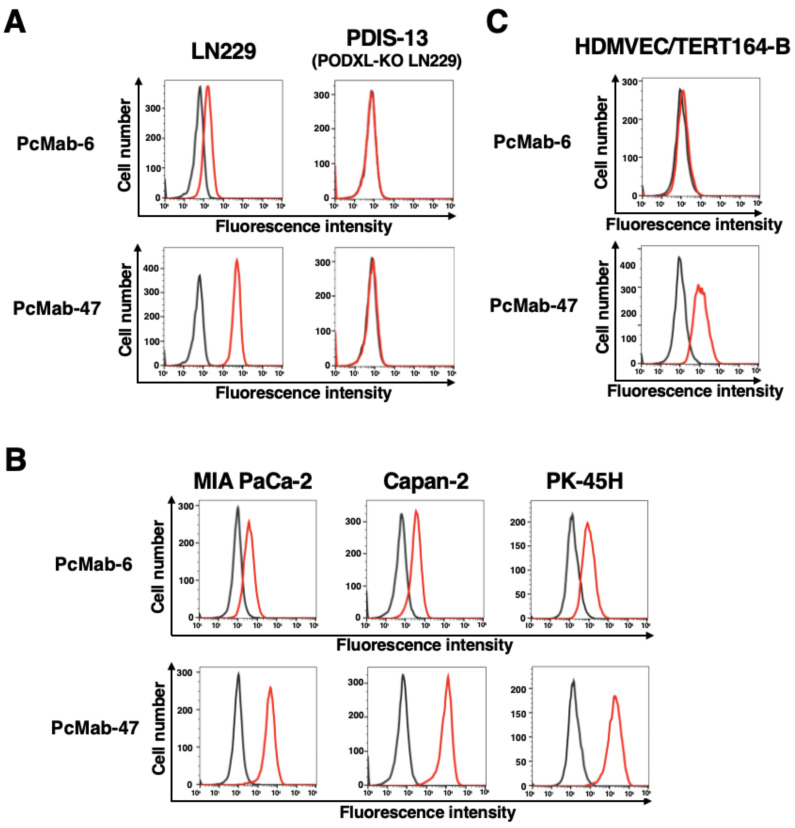
The reactivity of anti-PODXL mAbs (PcMab-6 and PcMab-47) against cancer and normal cells using flow cytometry. (**A**) LN229 and PODXL-KO LN229 (PDIS-13) cells were treated with 10 µg/mL of PcMab-6 (red line), PcMab-47 (red line), or blocking buffer (black line). (**B**) Pancreatic cancer MIA PaCa-2, Capan-2, and PK-45H cells were treated with 10 µg/mL of PcMab-6 (red line), PcMab-47 (red line), or blocking buffer (black line). (**C**) Lymphatic endothelial HDMVEC/TERT164-B cells were treated with 10 µg/mL of PcMab-6 (red line), PcMab-47 (red line), or blocking buffer (black line). Then, cells were treated with Alexa Fluor 488-conjugated anti-mouse IgG.

**Figure 2 ijms-25-00161-f002:**
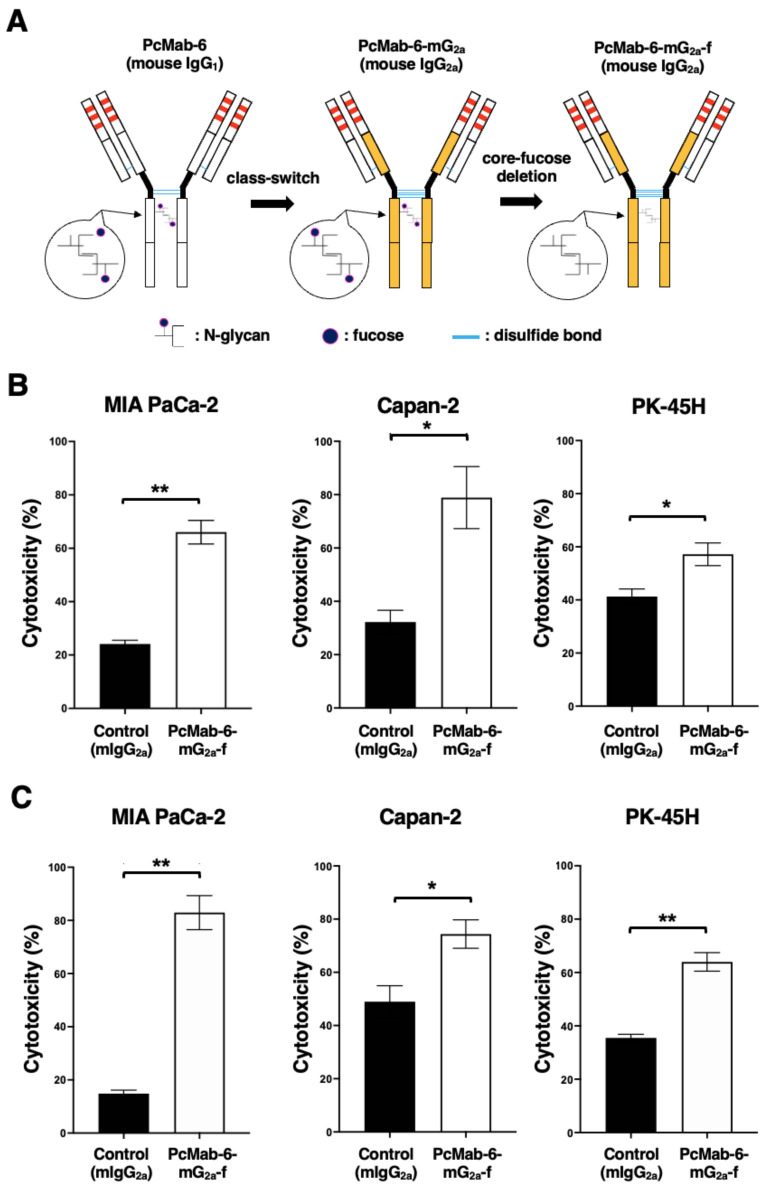
ADCC and CDC via PcMab-6-mG_2a_-f. (**A**) The V_H_ cDNA of PcMab-6 (mouse IgG_1_) and the C_H_ of mouse IgG_2a_ were cloned into a vector. The V_L_ and C_L_ cDNAs of PcMab-6 were cloned into another vector. To generate a core-fucose-deficient mouse IgG_2a_ mAb (PcMab-6-mG_2a_-f), the antibody expression vectors were transfected into BINDS-09 (Fut8-knocked-out ExpiCHO-S cells). (**B**) ADCC induced by PcMab-6-mG_2a_-f or control mouse IgG_2a_ (mIgG_2a_) against MIA PaCa-2, Capan-2, and PK-45H cells. (**C**) CDC induced by PcMab-6-mG_2a_-f or control mIgG_2a_ against MIA PaCa-2, Capan-2, and PK-45H cells. Values are shown as mean ± SEM. Asterisks indicate statistical significance (** *p* < 0.01, * *p* < 0.05; Welch’s *t*-test).

**Figure 3 ijms-25-00161-f003:**
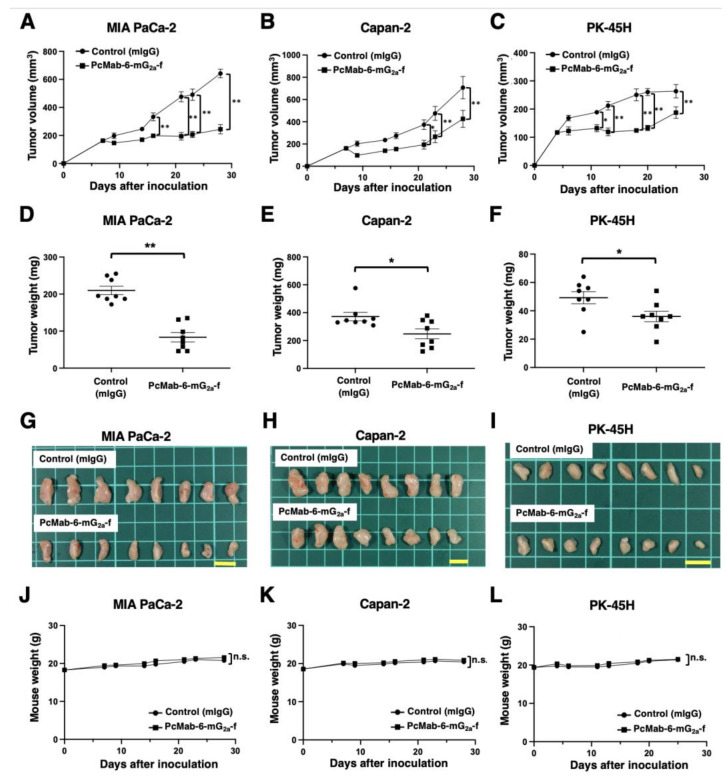
Antitumor activity of PcMab-6-mG_2a_-f against pancreatic cancer xenografts. (**A**–**C**) MIA PaCa-2 (**A**), Capan-2 (**B**), and PK-45H (**C**) cells were subcutaneously injected into BALB/c nude mice (day 0). In total, 100 μg of PcMab-6-mG_2a_-f or control mouse IgG (mIgG) were intraperitoneally injected into each mouse on day 4 (PK-45H) or day 7 (MIA PaCa-2 and Capan-2). Additional antibodies were injected on days 11 and 18 (PK-45H) or days 14 and 21 (MIA PaCa-2 and Capan-2). The tumor volume is represented as the mean ± SEM. ** *p* < 0.01, * *p* < 0.05 (ANOVA and Sidak’s multiple comparisons test). (**D**–**F**) The mice were euthanized on day 25 (PK-45H) or day 28 (MIA PaCa-2 and Capan-2) after cell implantation. The tumor weights of MIA PaCa-2 (**D**), Capan-2 (**E**), and PK-45H (**F**) xenografts were measured. Values are presented as the mean ± SEM. ** *p* < 0.01, * *p* < 0.05 (Welch’s *t*-test). (**G**–**I**) MIA PaCa-2 (**G**), Capan-2 (**H**), and PK-45H (**I**) xenograft tumors (scale bar, 1 cm). (**J**–**L**) Body weights of MIA PaCa-2 (**J**), Capan-2 (**K**), and PK-45H (**L**) xenograft-bearing mice treated with control mIgG or PcMab-6-mG_2a_-f. n.s., not significant.

**Figure 4 ijms-25-00161-f004:**
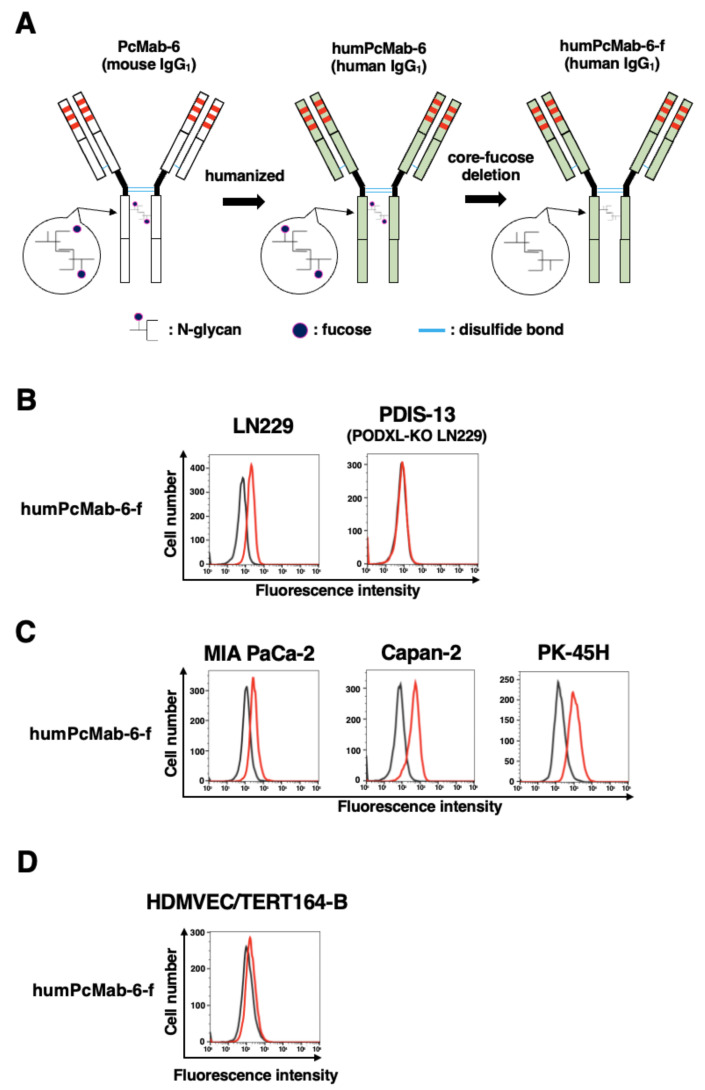
Establishment of the humanized and defucosylated antibody humPcMab-6-f. (**A**) The complementarity determining region (CDR) of PcMab-6 V_H_, the frame sequences of V_H_ in human Ig, and the C_H_ of human IgG_1_ were cloned into a vector. The CDR of PcMab-6 V_L_, the frame sequences of V_L_ in human Ig, and the C_L_ of the human kappa chain were cloned into another vector. To generate a core-fucose-deficient form (humPcMab-6-f), the antibody expression vectors were transfected into BINDS-09 cells. (**B**) LN229 and PODXL-KO LN229 (PDIS-13) cells were treated with 10 µg/mL of humPcMab-6-f (red line) or blocking buffer (black line). (**C**) MIA PaCa-2, Capan-2, and PK-45H cells were treated with 10 µg/mL of humPcMab-6-f (red line) or blocking buffer (black line). (**D**) HDMVEC/TERT164-B cells were treated with 10 µg/mL of humPcMab-6-f (red line) or blocking buffer (black line). Then, cells were treated with FITC-conjugated anti-human IgG.

**Figure 5 ijms-25-00161-f005:**
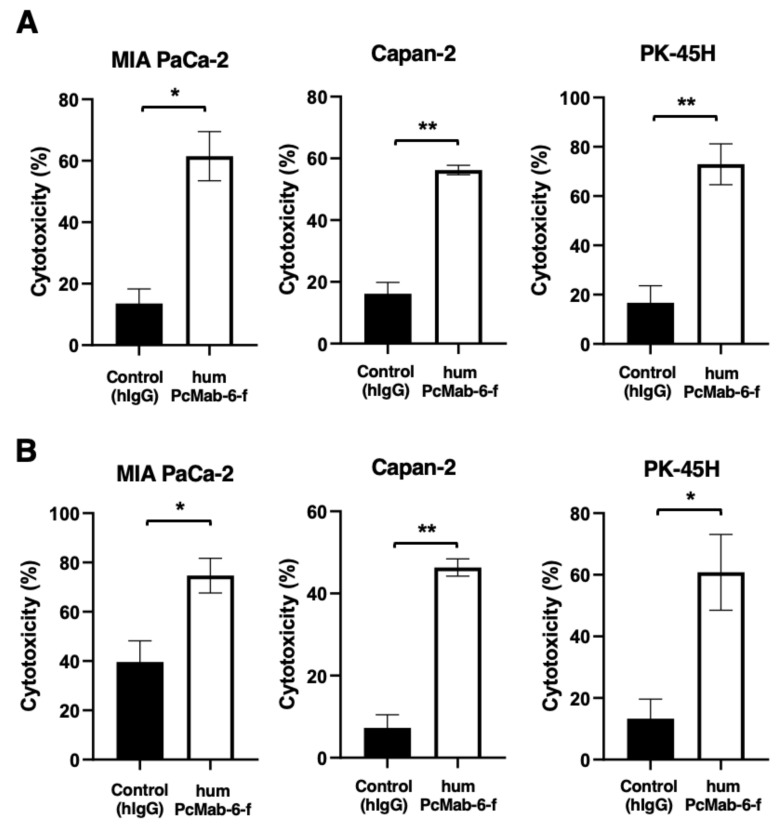
ADCC and CDC caused by humPcMab-6-f against pancreatic cancers. (**A**) ADCC induced by humPcMab-6-f or control human IgG (hIgG) against MIA PaCa-2, Capan-2, and PK-45H cells. (**B**) CDC induced by humPcMab-6-f or control hIgG against MIA PaCa-2, Capan-2, and PK-45H cells. Values are shown as mean ± SEM. Asterisks indicate statistical significance (** *p* < 0.01, * *p* < 0.05; Welch’s *t*-test).

**Figure 6 ijms-25-00161-f006:**
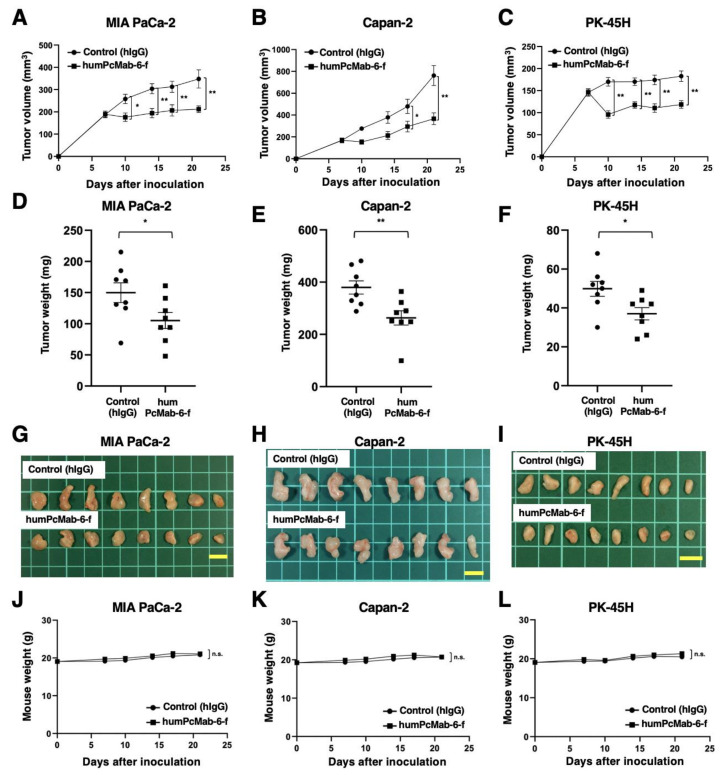
Antitumor activity of humPcMab-6-f against pancreatic cancer xenografts. (**A**–**C**) MIA PaCa-2 (**A**), Capan-2 (**B**), and PK-45H (**C**) cells were subcutaneously injected into BALB/c nude mice (day 0). In total, 100 μg of humPcMab-6-f or control hIgG were intraperitoneally injected into each mouse on day 7. Additional antibodies were injected on day 14. Human NK cells were also injected around the tumors on days 7 and 14. The tumor volume is represented as the mean ± SEM. ** *p* < 0.01, * *p* < 0.05 (ANOVA and Sidak’s multiple comparisons test). (**D**–**F**) The mice were euthanized on day 21 after cell implantation. The tumor weights of the MIA PaCa-2 (**D**), Capan-2 (**E**), and PK-45H (**F**) xenografts were measured. Values are presented as the mean ± SEM. ** *p* < 0.01, * *p* < 0.05 (Welch’s *t*-test). (**G**–**I**) MIA PaCa-2 (**G**), Capan-2 (**H**), and PK-45H (**I**) xenograft tumors (scale bar, 1 cm). (**J**–**L**) Body weights of MIA PaCa-2 (**J**), Capan-2 (**K**), and PK-45H (**L**) xenograft-bearing mice treated with control hIgG or humPcMab-6-f. n.s., not significant.

## Data Availability

The data presented in this study are available in the article and Appendix A.

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
