# Peer review of "A Cancer-Specific Monoclonal Antibody against Podocalyxin Exerted Antitumor Activities in Pancreatic Cancer Xenografts"

_ijms, 2023, doi:10.3390/ijms25010161_

Round 1

Reviewer 1 Report

Comments and Suggestions for Authors

1. Authors examined the antitumor activity of engineered antibodies against mouse xenograft models of PDAC. Why not use a PDAC isograft model? To what extent is the condition of xenograft involved in antitumor activity? Using naked mice is enough to prevent or decrease the response to xenograft?  To what extent is antitumor activity attributable to engineered antibodies?

2.. In this study, authors developed a cancer-specific anti-PODXL mAb, PcMab-6 (IgG1, kappa) by screening more than one hundred hybridoma clone, and engineered PcMab-6 into a mouse IgG2a type (PcMab-6-mG2a) and a humanized IgG1-type (humPcMab-6), and further produced the core fucose-deficient types (PcMab-6-mG2a-f and humPcMab-6-f, respectively) to potentiate the ADCC, and to reduce the risk of adverse effects on normal tissues.  However, how this technique reduces the risk of adverse effects or toxicities to normal vascular/lymphatic endothelial cells and kidney podocytes?

3. The authors could have performed general urine testing, blood count, and blood chemistry in the mice implanted with the tumor cell lines to rule out toxic effects of the antibodies used.

3. Figure 1 C. Lymphaic endothelial HDMVEC/TERT164-B cells were marked both with the buffer and by the Ab (PCMab-6)?

4. ADCC is higher than CDC, except against MIA PaCa-2. Why?

Author Response

  1. Authors examined the antitumor activity of engineered antibodies against mouse xenograft models of PDAC. Why not use a PDAC isograft model?

Unfortunately, PcMab-6 does not recognize mouse PODXL. Therefore, we cannot use any isograft models of PDAC.

To what extent is the condition of xenograft involved in antitumor activity? Using naked mice is enough to prevent or decrease the response to xenograft? To what extent is antitumor activity attributable to engineered antibodies?

In xenograft models in nude mice, NK cells and macrophages are thought to contribute to the ADCC-mediated antitumor effect. As shown in ADCC activity (Figure 2B), we used the splenocyte derived from nude mice and could observe the ADCC activity. Therefore, NK cells and macrophages would exert the antitumor effect in nude mice.

  1. In this study, authors developed a cancer-specific anti-PODXL mAb, PcMab-6 (IgG1, kappa) by screening more than one hundred hybridoma clone, and engineered PcMab-6 into a mouse IgG2a type (PcMab-6-mG2a) and a humanized IgG1-type (humPcMab-6), and further produced the core fucose-deficient types (PcMab-6-mG2a-f and humPcMab-6-f, respectively) to potentiate the ADCC, and to reduce the risk of adverse effects on normal tissues. However, how this technique reduces the risk of adverse effects or toxicities to normal vascular/lymphatic endothelial cells and kidney podocytes?

The technique of class-switching to IgG2a and core fucose deficiency enhances the antitumor effect if a mAb binds to the antigen. Due to the low reactivity to normal lymphatic endothelial cells (Figures 1C and 4D), adverse effects on normal tissues would be reduced. In fact, chPcMab-6 did not show any toxicities against cynomolgus monkeys in single intravenous injection (20 mg/kg) and 10 mg/kg in every week. We mentioned it in the discussion (Line 315).

  1. The authors could have performed general urine testing, blood count, and blood chemistry in the mice implanted with the tumor cell lines to rule out toxic effects of the antibodies used.

Since PcMab-6 does not recognize mouse PODXL, the mAb does not exhibit toxicity to murine normal cells. As mentioned above, we examined the toxicities in cynomolgus monkeys. The side effect was not observed, and the abnormality of urine, lung, and kidney was not detected.

  1. Figure 1 C. Lymphaic endothelial HDMVEC/TERT164-B cells were marked both with the buffer and by the Ab (PCMab-6)?

No. We measure the fluorescence at the same experimental setting of LN229 or PDAC cells. Since the cell size of HDMVEC/TERT164-B is larger than that of the above cell lines, the autofluorescence of HDMVEC/TERT164-B could influence the result.

  1. ADCC is higher than CDC, except against MIA PaCa-2. Why?

We are not sure of the reason. The sensitivity of ADCC assay is different in cell lines. MIA PaCa-2 may possess the mechanism to resist ADCC-mediated cellular toxicity.

Reviewer 2 Report

Comments and Suggestions for Authors

Although this manuscript has not performed many molecular level studies on why the discovered specificity differences of their mAb to cancer versus normal cell/tissue, the finding reported in this manuscript is very interesting and the data quality is good. The sutidy has potential important applications in cancer therapy and may also further increase the interests of this related research area for other targets to do the same. the following specific comments should be considered in the revised version of this manuscript before publication in IJMS.

1. Do these authors know why “PcMab-6 recognized pancreatic ductal adenocarcinoma (PDAC) cell lines (MIA PaCa-2, Capan-2, and PK-45H), but did not react with normal lymphatic endothelial cells (LECs). In contrast, one of the non-CasMabs, PcMab-26 showed high reactivity to both the PDAC cell lines and LECs.” This may be discussed in the Discussion section. Additionally, these authors should discuss in the Discussion section why “As shown in Figure 1C, PcMab-47 stained PODXL of a lymphatic endothelial cell (LEC), HDMVEC/TERT164-B. In contrast, PcMab-6 did not react with it (Figure 1C)”.

2. The Figure 1A showed the different reactivity intensity of PcMab-6 (weak) and PcMab-47 (very strong) in the PODXL-positive glioblastoma LN229 cells, which were further confirmed the same situation for 3 PDAC cell lines (Figure 1B). Same as Comment 1 above, these authors may discuss this in the Discussion section as well to balance the weakness of this manuscript with limited molecular mechanism studies.

3. The Figure 2A class-switch cartoon is not very clear to this Reviewer. More specifically, how the PcMab-6G1 was changed into PcMab-6mG2a? This reviewer saw the CH Change color and blue line bond linkers from 2 to 4. But not understand how the switch were managed. These authors need to revise their Figure 2A to let general audience to easily understand the switch.

4. While the data shown in Figure 2BC is very nice, a critical question for the data shown in Figure 2BC is that since these authors used cytotoxicity as an indicator, the so-called “Control mlgG2a” needs to be clearly defined. This reviewer’s understanding is that if the Control mlgG2a does not have the binding affinity to PODXL, the results shown in Figure 2BC would not reflect the specificity difference. Nevertheless, these authors need to define two related antibody difference and revise their writing to let general audience to understand how cytotoxicity difference could reflect ADCC and CDC?

5. Figure 4A has issue similar to Figure 2A from this reviewer (comment 3). The Switch to humanized process is not clear. Refer to Comment 3.   

6. Similar to the comment 4, while the data shown in Figure 5AB is very nice and clearly the control hIgG showed differences from the control mIgG2a in terms of cytotoxicity percentage, a critical question for the data shown in Figure 5AB is that since these authors used cytotoxicity as an indicator, the so-called “Control hIgG” needs to be defined as well. Similarly, if the Control hIgG does not have the binding affinity to PODXL, the results shown in Figure 5AB would not reflect the specificity difference. In other words, these authors need to define control hIgG antibody and revise the wording for general audience to be easier in understanding how cytotoxicity difference could reflect ADCC and CDC?

Author Response

Although this manuscript has not performed many molecular level studies on why the discovered specificity differences of their mAb to cancer versus normal cell/tissue, the finding reported in this manuscript is very interesting and the data quality is good. The sutdy has potential important applications in cancer therapy and may also further increase the interests of this related research area for other targets to do the same. the following specific comments should be considered in the revised version of this manuscript before publication in IJMS.

  1. Do these authors know why “PcMab-6 recognized pancreatic ductal adenocarcinoma (PDAC) cell lines (MIA PaCa-2, Capan-2, and PK-45H), but did not react with normal lymphatic endothelial cells (LECs). In contrast, one of the non-CasMabs, PcMab-26 showed high reactivity to both the PDAC cell lines and LECs.” This may be discussed in the Discussion section. Additionally, these authors should discuss in the Discussion section why “As shown in Figure 1C, PcMab-47 stained PODXL of a lymphatic endothelial cell (LEC), HDMVEC/TERT164-B. In contrast, PcMab-6 did not react with it (Figure 1C)”.

We answered with comment 2 (see below).

  1. The Figure 1A showed the different reactivity intensity of PcMab-6 (weak) and PcMab-47 (very strong) in the PODXL-positive glioblastoma LN229 cells, which were further confirmed the same situation for 3 PDAC cell lines (Figure 1B). Same as Comment 1 above, these authors may discuss this in the Discussion section as well to balance the weakness of this manuscript with limited molecular mechanism studies.

As we discussed in the discussion, PcMab-6 reacted with PODXL, which is overexpressed in glycan-deficient cell lines, such as Lec1, Lec2, and Lec8 (data not shown) in the same way with PcMab-47, suggesting that PcMab-6 might react with the glycan-independent conformational epitope of PODXL(Line 304).

We recently developed CasMabs (clone H2Mab-214 [36] and clone H2Mab-250 [37]) against HER2. These anti-HER2 CasMabsexhibited reactivity with cancer cells but no reactivity with normal cells. The structural analysis of H2Mab-214 revealed that H2Mab-214 recognizes the local misfolded cysteine-rich domain of HER2 (manuscript submitted [36]). Therefore, we described the discussion, as follows.

“Furthermore, the recognition by PcMab-6 in flow cytometry was lower than that of PcMab-47 (Figure 1), which can be attributed to the possibility that the epitope sequence is partially exposed or modified in cancer cells, but not in normal cells in living cells.” (Line 288)

Because ref 36 (preprint version) is under revision now, we did not discuss the detailed mechanism of the recognition by PcMab-6.

  1. The Figure 2A class-switch cartoon is not very clear to this Reviewer. More specifically, how the PcMab-6G1 was changed into PcMab-6mG2a? This reviewer saw the CHChange color and blue line bond linkers from 2 to 4. But not understand how the switch were managed. These authors need to revise their Figure 2A to let general audience to easily understand the switch.

To generate PcMab-6-mG2a, we first cloned the cDNA of the heavy and light chains of PcMab-6. The VH cDNA of PcMab-6 and CH of mouse IgG2a were subcloned into a vector, and VL and CL cDNAs of PcMab-6 were subcloned into another vector. To generate PcMab-6-mG2a-f, the antibody expression vectors were transfected into BINDS-09 (Fut8-knocked out ExpiCHO-S cells). We added the description in the legend of Figure 2.

The blue lines show the disulfate bonds. Mouse IgG1 has two, and mouse IgG2a has four disulfate bonds between the heavy chains.

  1. While the data shown in Figure 2BC is very nice, a critical question for the data shown in Figure 2BC is that since these authors used cytotoxicity as an indicator, the so-called “Control mlgG2a” needs to be clearly defined. This reviewer’s understanding is that if the Control mlgG2a does not have the binding affinity to PODXL, the results shown in Figure 2BC would not reflect the specificity difference. Nevertheless, these authors need to define two related antibody difference and revise their writing to let general audience to understand how cytotoxicity difference could reflect ADCC and CDC?

Control mlgG2a does not recognize PODXL. We described this in the result section (Lines 113 and 120).

  1. Figure 4A has issue similar to Figure 2A from this reviewer (comment 3). The Switch to humanized process is not clear. Refer to Comment 3.

To generate a humanized PcMab-6 (humPcMab-6), the complementarity determining region (CDR) of PcMab-6 VH, frame sequences of VH in human Ig, and CH of human IgG1 were cloned into a vector. The CDR of PcMab-6 VL, frame sequences of VL in human Ig, and CL of human kappa chain were cloned into another vector. To generate humPcMab-6-f, antibody expression vectors were also transfected into BINDS-09 (Fut8-knocked out ExpiCHO-S cells). We added the description in the legend of Figure 4.

  1. Similar to the comment 4, while the data shown in Figure 5AB is very nice and clearly the control hIgG showed differences from the control mIgG2a in terms of cytotoxicity percentage, a critical question for the data shown in Figure 5AB is that since these authors used cytotoxicity as an indicator, the so-called “Control hIgG” needs to be defined as well. Similarly, if the Control hIgG does not have the binding affinity to PODXL, the results shown in Figure 5AB would not reflect the specificity difference. In other words, these authors need to define control hIgG antibody and revise the wording for general audience to be easier in understanding how cytotoxicity difference could reflect ADCC and CDC?                                                                                     

Control hlgG does not recognize PODXL. We described this in the result section. (Lines 195 and 202)